# Emerging Designs of Electronic Devices in Biomedicine

**DOI:** 10.3390/mi11020123

**Published:** 2020-01-22

**Authors:** Maria Laura Coluccio, Salvatore A. Pullano, Marco Flavio Michele Vismara, Nicola Coppedè, Gerardo Perozziello, Patrizio Candeloro, Francesco Gentile, Natalia Malara

**Affiliations:** 1Department of Experimental and Clinical Medicine, University of Magna Graecia, 88100 Catanzaro, Italy; coluccio@unicz.it (M.L.C.); gerardo.perozziello@unicz.it (G.P.); patrizio.candeloro@unicz.it (P.C.); 2Department of Health Sciences, University of Magna Graecia, 88100 Catanzaro, Italy; pullano@unicz.it (S.A.P.); marco@vismara.info (M.F.M.V.); 3Institute of Materials for Electronics and Magnetism (IMEM) CNR Parco area delle Scienze 37/A, 43124 Parma, Italy; nicola.coppede@gmail.com; 4Department of Electrical Engineering and Information Technology, University Federico II of Naples, 80125 Naples, Italy

**Keywords:** biodevices, integration, miniaturized devices

## Abstract

A long-standing goal of nanoelectronics is the development of integrated systems to be used in medicine as sensor, therapeutic, or theranostic devices. In this review, we examine the phenomena of transport and the interaction between electro-active charges and the material at the nanoscale. We then demonstrate how these mechanisms can be exploited to design and fabricate devices for applications in biomedicine and bioengineering. Specifically, we present and discuss electrochemical devices based on the interaction between ions and conductive polymers, such as organic electrochemical transistors (OFETs), electrolyte gated field-effect transistors (FETs), fin field-effect transistor (FinFETs), tunnelling field-effect transistors (TFETs), electrochemical lab-on-chips (LOCs). For these systems, we comment on their use in medicine.

## 1. Introduction

Theranostics is universally understood to be the use of a combination of nanoscale agents and techniques that have both diagnostic and therapeutic effects on a disease. Theranostics provides a transition from conventional medicine to a personalized and precision medicine approach.

It includes a large variety of themes including, for example, molecular imaging, personalized medicine, or pharmacogenomics that expand the field of knowledge on targeted therapies and enhance our understanding of the molecular mechanisms of drugs. In the last years, recent advances in microfluidics [1,2,3,4] and nanotechnology [5,6,7,8] gave significant support to theranostics for developing new procedures and treatments of diseases.

This review focuses on the development of strategies and potential applications of emerging theranostic nanosystems based on the transport of electroactive species (i.e., ions or electrons) at the nanoscale. Faster diagnosis and screening of diseases have become increasingly important in predictive personalized medicine as they improve patient treatment strategies and reduce cost as well as the burden of healthcare. However, the correct extraction, acquisition, and sampling of physiological signals is still unsatisfied by many of the currently available approaches; for these, the analysis of body fluids such as tears, sweat, saliva, or interstitial fluid, is heavily conditioned by the type of biomarker integrated with the electrochemical/bioelectronics sensor that performs the analysis.

The choice of the correct biomarker is a critical step that can compromise the entire process of measurement and overrule even the more advanced technology of a sensing device. The blind biomarker should exhibit the properties of high specificity and sensitivity in monitoring a disease. This adherence must be investigated in a preliminary stage, before the use of the biomarker in the device in its final configuration. Then, the aim of the (electrochemical) device is that of enhancing the sensing abilities of the biomarker in terms of limit of detection (i.e., the smallest quantity or concentration of the analyte to be detected), resolution (i.e., the smallest incremental unit of the analyte that the biosensor can detect), and sensitivity (i.e., how much the response of the system changes as the input changes). A nanoscale architecture of the device can improve the limit of detection, resolution, and sensitivity of the biosensor. This review is an account of how nanotechnology can enter the field of electrochemical transistors to impact clinical medicine. The circuit functionalities and their applications will also be addressed, with attention to current trends in the field.

For their characteristics of high sensitivity and fast response times, electrochemical biosensors may provide early diagnosis of diseases and increase the possibility of a patient’s recovery. Perhaps more importantly, electrochemical biosensors are able to translate a chemical signal into an electrical signal and this enables us to detect and quantify several different molecular or cellular species in the body. Moreover, these systems can be integrated with lab-on-chips to obtain point of care (POC) analytical platforms.

In the fields of theranostics and prognosis, it is necessary to develop devices with a high impact on the detection sensitivity and specificity of the biomarker that in turn must be specific and adherent to the disease under examination. The integration between electrochemical biosensors and lab-on-chips (LOCs) to obtain POC analytical platforms is discussed with plenty of examples. We report, describe, and comment on latest generation transistors, electrochemical biosensors (fin field-effect transistors (FinFETs), tunnelling field-effect transistors (TFETs), and organic electrochemical transistors (OECTs)), and the combination of electrochemical biosensors with lab-on-chips for medical applications.

## 2. Fin Field-Effect Transistor (FinFET), Tunnel FET

The field-effect transistor is a type of transistor that uses an electric field to control the flow of current. A typical FET device has three terminals: source, gate, and drain. The application of a voltage to the FET’s gate modifies the conductivity between the drain and the source, allowing the control of the flow of current.

One of the first examples of field-effect devices for the evaluation of ionic species was introduced by Bergveld in the 1970s and named ion sensitive field-effect transistor (ISFET) [9].

In this class of devices, the gate consists of an SiO_2_ layer placed in solution, and consequently, the drain current is influenced by analyte activity by varying the potential at the gate/electrolyte interface [9,10]. Subsequently, different technologies of field-effect devices were developed to overcome the main limits imposed by FET–based devices, such as the threshold voltage drift. In chronological order the extended gate field effect transistor (EGFET) was developed by Van der Spiegel in the early 1980s [11]. The continuous scaling of planar metal-oxide-semiconductor field-effect transistor (MOSFET) evidenced come technological difficulties whereby the device can no longer be classified as a long channel MOSFET. The main short-channel effect is due to the two-dimensional distribution of potential and high electric fields in the channel region, which mainly lead to variable threshold voltage, saturation region that does not depend on drain potential, and drain current that does not depend on the inverse of channel length [12]. Subsequently a variety of different planar and non-planar topologies were investigated, such as the FinFET and the TFET [13,14]. The FinFET resulted in an attractive option for the fabrication of a non-planar device with self-aligned double-gate using a standard complementary metal-oxide semiconductor (CMOS) process [15]. Conversely, the FET is considered a promising design which guarantees immunity to subthreshold swing degradation at short channel length [16]. More recently, the development of more sophisticated organic materials led to the development of devices that are trying to replace the role of silicon, such as the organic thin-film transistors (OTFTs). Even though the OTFTs’ performances are actually not comparable with inorganic ones in terms of carrier mobility, operating frequency, and subthreshold swing, the low cost and easier fabrication, as well as the possibility to chemically modify the material properties, represent a key role in the development of organic biosensors [17].

The electronic interface of a field-effect biosensors’ electronic interface is a standard and/or custom designed MOSFET, which ensures long-term stability and insulation from the chemical environment to the device (where the sensing layer is generally placed) [16]. In linear region, the sensor output (i.e., drain current) is related to the analyte as follows:(1)$$I_{DS}=\mu C_{ox}\frac{W}{L}\left[\left(V_{Ref}-V_{th}^{*}\right)V_{DS}-\frac{1}{2}V_{DS}^{2}\right]$$
here W is the width and L is the length of the channel, *μ* is the carrier mobility, $C_{Ox}$ is the gate oxide capacitance per unit area, and $V_{Ref}$ and $V_{DS}$ are the applied reference electrode and the drain-to-source voltages. $V_{th}^{*}$ is the threshold voltage, which is related to the device and the chemical environment as follows: (2)$$V_{th}^{*}=V_{th}+E_{ref}+\chi_{sol}-\frac{W_{M}}{q}-\phi$$

In Equation (2), $V_{th}$ is the threshold voltage of the field-effect device. $E_{Ref}$ is the reference electrode potential, $\chi_{sol}$ is the dipole potential of the electrolyte, $W_{M}$ is the work function of the reference electrode, q is the charge, and φ is the potential at the sensing interface [10]. In Figure 1 is reported a comparison between an EGFET and a Fin-FET device for biosensing.

In the EGFET aspect ratio W/L influences the characteristics of the devices in terms of transconductance $g_{m}$. A higher transconductance is desirable because it results in lower flicker (1/f) noise device. There are several advantages in FinFET such as a lower off current, a lower $V_{th}$ due to a reduced bulk (depletion) capacitance and a very low output conductance (higher voltage gain). Conversely, FinFETs suffer from a high series resistance and thus a lower peak transconductance [18]. In both cases the literature reports that most of the developed biosensors are developed using commercial devices, evidencing how the development of biosensors over the years has been mostly oriented toward the sensing part, using off-the-shelf components because of the easier fabrication process and lower cost [19,20]. Even though they are a more recent technology, FinFETs have more recently been commercialized and thus are mature for biosensor applications [21]. One of the key metrics in the development of biosensors is their sensitivity, which through the years has been the object of intensive investigation, especially for the detection of analytes at even lower concentration. Being inherently characterized by theoretical limits other approaches were investigated instead of classical planar and non-planar geometries. The TFET is one of the most recent devices, with base conduction mechanism on the band-to-band tunneling. In this class of device, the analyte influences the tunneling barrier, and hence the tunneling current. Literature has evidenced that the use of TFET technology results in devices with improved sensitivity and reduced response time, while retaining all other advantages of FET biosensors [22,23,24]. FinFET technology was originally proposed as an improved technology characterized by higher sensitivity, stability, and reliability [25]. Literature reports some attempts to fabricate FinFET-based sensors for biomolecule detection such as cellular ion activities [26], pH [25,27], and the detection of avian influenza (AI) antibody [28]. Change in current was recently linked to change in gate capacitance, allowing the detection of proteins linked to early detection of diseases (e.g., streptavidin, biotin) [29]. Moreover, being a relatively recent technology, modelling tools are still under development to optimize the design phase [30]. 

The continuous efforts to improve the sensing performances attracted significant attention through the recent technological advancement in synthesis and deposition on high performance materials, such as graphene (i.e., nanopores, nanoribbon, reduced graphene oxide and graphene oxide), carbon nanotube, nanowires, and nanoporous materials [31,32,33,34,35,36,37,38]. Graphene is a high-performance material, recently investigated in different fields due to the availability of synthesis and mature deposition technologies, characterized by high carrier mobility and low inherent 1/f noise [39]. As result, different attempts at using a graphene-FET (GFET) as biosensor were reported in literature. Most of the applications are focused on low-concentration nucleic acid detection, exploiting the site-specific immobilization of probes [32]. The reported resolution of GFETs, often conjugated with metal nanoparticles (e.g., Au) can be lowered down to the pM range [40,41]. Despite the interesting sensing applications, continuous investigations are still required to improve the reduced DNA translocation dynamics and the low-frequency noise levels [32,42,43]. Other applications are concerned with protein detection, living cell and bacteria monitoring [33,44]. A commercial graphene-based biosensor is the agile biosensor chip—NTA, used overall for research purposes, allowing the immobilization of recombinant proteins.

The efforts to develop the FinFET device often lead to biosensors with performances comparable with that of other multigate or planar MOSFETs [25]. Subsequently, in order to reduce the development time, commercial FET devices are sometimes preferred.

## 3. Organic Electrochemical Transistor Devices 

The role of organic electrochemical transistors in biomedicine is becoming increasingly relevant. OECTs are devices based on a semiconductor, conventionally named channel, that is physically placed in contact with a solution. Upon the action of an externally controlled voltage, ions of the solution are displaced and can be either injected or removed from the channel, doping or dedoping it, changing the bulk conductivity of the entire device. Alterations in the electrical characteristics of the device can be in turn correlated to the physical and chemical characteristics of the electrolyte. State of the art OECTs are mostly based on the conducting polymer poly(3,4-ethylenedioxythiophene) doped with polystyrene sulfonate (PEDOT:PSS) [45], thus OECTs share many of the characteristics of polymers such as low weight, low density, low cost, high resilience, elevated specific strength, biocompatibility, and facile deposition. Moreover, because they have a transistor architecture, OECTs feature high sensitivity, high signal-to-noise ratio, high gain [46,47,48], enabling amplification of weak electric signals such as those generated by biological systems. 

OECTs, in fact, can operate at low voltages, in aqueous environments such as efficient ion-to-electron converters, providing an interface between the worlds of biology and electronics. 

As key constituents of biosensors and analytical devices, OECTs have been used for electrophysiological recording, for bio-sensing applications and applications at the bio-interface [45,49,50], bio-computing [51], neuromorphic engineering [52,53,54], as constituents in electronic bio-devices [55], and as a sensor for cells [56].

As flexible, high-sensitivity low-cost devices, OECTs have found different applications in biomedicine and biology. They have been applied as sensors for simple analytes such as hydrogen peroxide [57] and ions [47,58]. In reference [59] Seong-Min Kim and colleagues examined the long term stability of PEDOT:PSS, examined the correlations among the microstructure, composition, and device performance of PEDOT:PSS films for possible applications in the development of long-term stable implantable bioelectronics for neural recording/stimulation. In reference [60], devices based on the conducting polymer PEDOT:PSS have been demonstrated for the real-time processing and manipulation of signals from living organisms; devices with the characteristics of miniaturization and bio-compatibility with human skin have been used to analyze neurophysiological activity. In references [61,62,63] it is discussed how similar OECTs can be used as sensing interfaces of cells and systems of cells. OECTs can very practically determine the physiological conditions of living cells, follow the processes at the basis of their life cycle, including reproduction processes, and track their transition to apoptosis [63].

Moreover, OECTs can be used as electronic switches or components of logic gates, also in interaction with biological interfaces, creating a multiple interactive logic that is perfectly suited to talk with living systems [64]. The in vivo monitoring of biological-driven phenomena is likewise heavily investigated, including, for example, enzymatic interactions [65] useful to detect metabolites relevant in in the biological processes and function of cells and organs, such as lactate or glucose, which are indicative of the physiological conditions of a patient [66]. Finally, a wide range of applications is still open in the detection of biomolecules based on specific antigens, which are crucial in the in vivo early diagnosis of bacteria and specific illnesses [67].

The typical configuration of an OECT is reported in Figure 2. An electrolyte, i.e., a solution containing electroactive species, is contacted to the device with three electrodes: the gate, the drain, and the source. The gate is the reference electrode that is directly connected to the electrolyte. Instead, the drain and the source are bridged together by the conductive polymer channel (Figure 2a). Upon application of a voltage between the gate and the source ($V_{gs}$) and the drain and the source ($V_{ds}$), ions in the electrolyte are propelled towards the polymer channel, penetrate into the channel, and generate a current $I_{ds}$ that flows from the drain to the source (Figure 2b). Thus, the current $I_{ds}$ is the typical output of an OECT device. The reaction between the ions and the polymer in the channel is described by the following equation:(3)$${\mathrm{PEDOT}}^{+}:{\mathrm{PSS}}^{-}+M^{+}+e^{-}\to \mathrm{PEDOT}+M^{+}:{\mathrm{PSS}}^{-}$$
where $M^{+}$ represents the cations. The presence of cations in the PEDOT:PSS film depletes the number of available carriers, reducing the source–drain current $I_{ds}$. Thus, $I_{ds}$ is modulated by the inflow or outflow of ions—from the electrolyte to the channel—and values of current measured by the device can be indicative of the concentration, size, and charge of the specie initially dispersed in solution, and of the geometrical characteristics of the system. The form of the $I_{ds}$ current is similar to response of a first order system: values of current are a function of time and increase from a reference value (minimum background current, no flux) to a steady state value (maximum value of current, constant regime) (Figure 2c). The signal can be modelled by an exponential function of the type $I_{ds}\sim m\left(1-e^{-t/\tau }\right)$, where t is time, and m and τ are the modulation and time constants of the system. The modulation is the signal increment normalized to its initial value, $m=\left(I_{ds}^{fin}-I_{ds}^{0}\right)/I_{ds}^{0}$; it is proportional to the strength of the signal. The time constant is the time after which the signal attains 67% of its steady state value, $I_{ds}\left(\tau \right)\sim 0.67\;I_{ds}^{fin}$; it is indicative of the inertia of the system. Remarkably, using mathematical models described elsewhere [68] m and τ can be associated with the diffusivity, charge, concentration, and other physical, chemical and geometrical characteristics of a system, so that the information encoded in the $I_{ds}$ can be broken to extract the characteristics of the system in analysis. Typical $I_{ds}$ values and increments fall in the 0–5 mA range. The values of $V_{ds}$, instead, are controlled by the operator and are typically varied in discrete increments in the 0–1 V range.

In this scheme, the electrolyte is contained in a channel, a chamber, or a reservoir; the interface of the solution with the external regions of the device is a flat surface with zero curvature. This automatically implies that the motion of ions in the system to the active sites of the device is driven by diffusion: the process is inefficiently controllable by the outside. By nanostructuring of the surface of the polymer channel, one can make the surface super-hydrophobic. Super-hydrophobicity prevents wetting of the surface, allowing a drop of solute positioned on that surface to maintain its originating spherical shape. The curvature of the drop can be modulated by tailoring the geometry of the super-hydrophobic surface and by regulating the amount of the solute partitioned in each drop. Thanks to a non-zero curvature, convective Marangoni flows arise in the drop (Figure 3a).

Depending on the value of curvature, the size of the drop, and the gradient of temperature between the substrate and the drop, the intensity of the convective fields can be equal, or greater or less, than the intensity of diffusion. Thus, convection represents the additional degree of freedom introduced in the system [69], and the competition between convection and diffusion drives the solute species on predefined spots. The velocity field developed within the drop owing to Marangoni flows is derived as the derivative of the stream functions ψ(r,θ) with respect to the coordinates r and θ. The stream functions are potential functions that describe the characteristics of a fluid flow, they have been originally derived by Tam and collaborators and [70] read:(4)$$\psi \left(r,\theta \right)=-\frac{1}{8}\left(1-r^{2}\right)\left[1+r\;cos\theta -\frac{1-r^{2}}{{\left(r^{2}+1-2r\;cos\theta \right)}^{\frac{1}{2}}}+\sum_{n=2}^{\infty }\frac{\left(n-1\right)-2\left(n-1\right)Bi}{\left(2n-1\right)\left(\left(n-1\right)+Bi\right)}r^{n}\left(P_{n-2}\;cos\theta -P_{n}\;cos\theta \right)\right],$$
where (r,θ) is the position of a point in the drop in polar coordinates, Bi is the Biot number, and $P_{n}$ is the Legendre polynomial of order n. The corresponding velocity field v is then derived as:(5)$$v_{r}=-\frac{1}{r^{2}sin\theta }\frac{\partial \psi }{\partial \theta },\;v_{\theta }=\frac{1}{r\;sin\theta }\frac{\partial \psi }{\partial r},$$
that can be used, in turn, in the Langevin equation [71,72,73] to find the distribution of a trace in the drop:(6)$$m\frac{\partial u}{\partial t}=6\pi \mu a\left(K_{p}u-K_{f}v\right)+F_{e}+F_{b}.$$

In Equation (6) u is the unknown velocity vector for the particle, v is the unperturbed fluid velocity, m and a are the mass and radius of the solute particulates. Moreover, $F_{e}$ is the electrostatic force, $F_{b}$ is the Brownian force that depends on the temperature as $F_{b}\propto \sqrt{T}$, $K_{p}$ and $K_{f}$ account for the hydrodynamic hindrance of the system, and t is time. Equation (6), solved using a numerical scheme, allows us to determine how a solute propagates in a super-hydrophobic drop because of convection and diffusion.

Inspired by Lotus leaves and by nature, super-hydrophobic surfaces have been reproduced using combinations of nano-fabrication techniques [74,75]. Typically, the artificial analogue of a super-hydrophobic surface is an array of microsized pillars, where the size (d), spacing (δ), and height (h) of the pillars in the array can vary over large intervals. The upper surface of the pillars contains, in turn, details at the nanoscale, and the combined effects across length-scales cause the surface to be super-hydrophobic [76,77,78]. The pillars of those surfaces are often made out of silicon, nano-machined using reactive ion etching techniques, or of polymers, created using optical or electron beam lithography techniques [79]. Fluorinated polymers with low friction coefficients, nano-porous silicon, or nano-rough materials with low surface energy densities, can be deposited on the pillars representing the second-level roughness of the hierarchical nanomaterial device [80]. The contact angle of a drop on similar super-hydrophobic surfaces can be predicted using the celebrated model of Cassie and Baxter [74]:(7)$$cos\left(\vartheta^{c}\right)=-1+\phi \;cos\left(\vartheta \right),$$
where ϑ is the contact angle of the drop on the surface without texture, $\vartheta^{c}$ is the contact angle on the surface with the texture, and ϕ is the solid fraction of the surface. Typical design values of d and δ are d=10 μm and δ=20 μm, so that $\phi =\pi /4{\left(d/\delta \right)}^{2}\sim 0.087$. With these values of d, δ and ϕ, any originating contact angle ϑ>60° will lead to final contact angles $\vartheta^{c}>150{}^{\circ}$, i.e., a super-hydrophobic surface. For this combination of d and δ, the height of the pillars should be chosen such that h≥20 μm, in that ratio of h to δ is greater than one, h/δ>1, to assure stable adhesion of the drop on the surface [81].

Motivated by the need of new sensor devices with higher sensitivity, higher accuracy, and increased selectivity with respect to available approaches, beginning in 2014 some of the authors of this paper started to nanostructure OECTs with the aim to harness their functionalities [82]. Starting from conventional OECT devices, we modified the geometry of those devices at the nanoscale, and created a new class of bio-devices that we called surface enhanced organic electrochemical transistors (SeOECTs) [69]. SeOECTs are a third generation of organic thin film transistors, in which the electrolyte medium is an active part of the device gating and the surface micro and nanostructure enhances the properties of the electrochemically active conductive polymer. In SeOECTs, a 3-dimensional design and topographical modification of the surface enables selectivity, enhances sensitivity, and enables the detection of multiple analytes in very low abundance ranges.

SeOECTs are based on the fine tailoring of surface microstructure and nano structure. The device comprises arrays of super-hydrophobic micro-pillars, functionalized with a conductive PEDOT:PSS polymer sensitive to the ionic strength of the electrolyte. Each pillar has a diameter of 10 μm and a height of 20 μm. The pillars are positioned on the substrate to form a non-periodic lattice (Figure 4a). A similar non-uniform tiling of pillars generates a system of radial forces that recalls the drop to the center of the lattice for automatic sample positioning. Some of the pillars are individually contacted to an external electrical probe station for site selective measurement on the sample surface (Figure 4b,c); they incorporate nano-gold contacts with sub-micron reciprocal distance that generate enhanced and localized electric fields (Figure 4d–h). Due to the microstructure of the device, the device is super-hydrophobic with contact angles up to 165° (Figure 4i). The device takes advantage of a combination of scales to resolve, identify, and measure complex biological mixtures. At the micro-scale, arrays of super-hydrophobic micro pillars enable manipulation and control of biological fluids. At the nano-scale, some pillars are modified to incorporate nano-electrodes for time and space resolved analysis of solutions.

SeOECTs are obtained by the superposition of different layers as explained in detail in reference [69].

To perform a measurement, a liquid sample is positioned on the device. Due to the super-hydrophobic characteristics of the surface, the sample takes a quasi-spherical shape (drop) and is automatically centered on the device. Then, the device is driven by an externally applied voltage, ranging between 0 and 1 volt. Upon the application of the voltage, a current of ions $I_{\mathrm{ds}}$ flows from the sample through the circuit and is measured by the points of measurements (sensors) on the device. Ions in the sample drop are transported by buoyancy and Marangoni flows that originate in the drop because of its curvature. Since the motion of ions—under the combined effect of convective flows and electric field—is directly proportional to the charge, directly proportional to the diffusion coefficient, and inversely proportional to the size, the process achieves the migration and spatial separation of species in solution. Arrays of sensors, spatially positioned on the device, can resolve this separation in space and time. Thus, the device measures ionic current transients at different positions on the substrate and for different values of voltage. While the information content of the solution is mapped into a whole set of variables, statistical techniques of analyses can be used to decode such information and determine the characteristics of target molecules.

The state of the system in a specific configuration is a point in the m−τ plane. Samples with different characteristics are placed in different regions of the diagram (Figure 5a). Thanks to this graphical representation, it is possible to operate sample separation, clustering, and classification (Figure 5b). The separation can be optimized if one considers sensors positioned at opposite extremes of the device or high values of voltage, for which the convective transport effects are amplified and the differences between species with different charge and size are maximized (Figure 5c). Points in the m−τ diagram can be parametrized by voltage (Figure 5d) or by time (Figure 5e). In both cases, data processing and analysis generate trajectories, the shapes of which are indicative of the time evolution and of the characteristics of the system (Figure 5f).

SeOECT devices have been used to evaluate tumors [83]. Since cancerous states are associated with an altered protonation state of the intra/extracellular microenvironment, one can estimate the onset and progression of a cancerous disease from a measurement of the ionic content of a blood-derived cell culture. We used SeOECTS to evaluate potential perturbation of protein protonation state (i.e., charge) of cell secretome in the extracellular compartment in vitro. We applied the analysis to the conditioned medium of blood culture after a short-time expansion, derived from patients with, without, and suspected of cancer. Using data from the bio-chip and statistical techniques of analysis, we developed algorithms that segregated tumor patients from non-tumor patients. For the ~30 patients across two independent cohorts, the method identified tumor patients with high sensitivity and 93% specificity [83].

## 4. Combined Electrochemical Biosensor and Lab-on-Chip

The sensing platform is then a crucial point for a specific application of LOCs, consisting of a sort of recognition element required for the capture of the target. The high affinity between antigen–antibody (Ag–Ab) or protein–aptamer makes them largely applied for biosensing design. Nanotechnologies offer support as electrochemical devices because the nanomaterials employed, from silicon to graphene or graphene oxide to carbon nanotubes or to metal nanoparticles, are characterized by excellent electrical, mechanical, and, generically, physical/chemical properties, which, combined with the novel nanotechnology techniques (lithography, metal deposition, plasma treating), allow the realization of appropriate architectures with the appropriate functionalities too. In particular the optical and electrical properties of the sensing nanodevices are related to their materials as well as to their geometry, and an optimal combination of them can amplify the signals coming from the analytes. Consequently, high-sensitivity analyses of biomarkers (usually cellular biomarkers or biomolecules) become possible adopting nanodevices with different techniques (e.g., cyclic voltammetry (CV), electrochemical impedance spectroscopy (EIS), IR, or Raman spectroscopy) [84,85]. Reference [7] reports one of the first examples of a sensor able to capture circulating tumor cells (CTCs), which could be used to study tumor staging, to guide the study of the risk of recurrence. Folic acid (FA) was selected as transducer molecule, because CTCs characterized by high expression of folate receptor (FR) and, consequently, evidencing also 5-methylcytosine-positive nuclei, are potentially dangerous for their dissemination power in healthy tissue. The FA surface can trap cancer-cell-expressing FR, encouraging the attack and the growth of CTC subsets with a higher content of methylated genomic DNA, with notable consequences on tumor prevention strategies and on prognosis definition.

Microfluidic paper-based analytical devices (μPADs) represent a technology of hydrophilic/hydrophobic micro-channel networks and associated analytical devices for development of portable and low-cost diagnostic tools that improve point of care testing (POCT) and disease screening [86] involving the specific detection of biomolecules. They have the ability to perform laboratory operations on micro-scale, using miniaturized equipment, and can be fabricated by using 2-D [86,87,88] or 3-D [89,90] methods to transport fluids in both horizontal and vertical dimensions, depending on complexity of the diagnostic application. The principal techniques in the literature for fabrication of paper-based microfluidic devices include: wax printing [90], inkjet printing [91], photolithography [92], flexographic printing [93], plasma treatment [94], laser treatment [95], wet etching [96], screen printing [97], and wax screen printing [98]. The μPADs can be used with the naked eye for qualitative testing but can also be used as quantitative assays based on specific detection methods. The choice of the method to detect the binding events that occur on a transducer surface depends on the type of biomarker. The detection methods currently used for sensitive measurements with low detection limits are represented by colorimetry, electrochemistry, fluorescence, chemiluminescence (CL), electrochemiluninescence (ECL), and photoelectrochemistry (PEC). Moreover, the surfaces can be modified to impart high selectivity to the binding of the target analyte, which is desirable in complex biological samples. In fact, sensitivity and specificity of the μPADs can be enhanced through a combination use of the reaction mechanisms categorized into biochemical, immunological, and molecular detections, transforming the device in a multiplexed testing [99,100,101]. However, chemical amplification or multiplex procedures are often disadvantaged in the μPADs as they require expensive reagents, multiple steps that must be performed by the end user, and complex protocols for the interpretation of the final data.

## 5. Future Trends of Combined Electrochemical Biosensor and Lab-on-Chip

The μPADs promise to meet the critical needs of rapid analytical tests in the diagnostic area. These devices represent the diagnostic field, a platform for a wide variety of chemical and biochemical reactions and detection patterns that can be used to assess the health status of the general population. They are useful for people who must reach very distant health facilities (Figure 6) [102]. These devices can also be used to monitor intoxications in occupational medicine. Finally, their ever-increasing application in the qualitative assessment of foods is beginning. Further research is needed to address several common challenges, such as the poor reproducibility, the need of high detection limits, the inadequate specificity, and the risk of a subjective interpretation of data. Most μPADs successfully address most of these challenges through the association of a machine learning procedure based on “kernel machines.” Kernel machines have considerable appeal in the machine learning research community due to a combination of conceptual elegance, mathematical tractability, and state-of-the-art performance [102], and, applied on Android smartphones for image processing and paper-based devices, they are able to solve many of the several common challenges mentioned.

The calculations of the limit of detection (LOD) and limit of concentration (LOC) for the combination of Android smartphones with image processing and paper-based devices are considered the frontier for their use. Despite its limitations, the interesting combination of the μPAD and Android applications provides a coherent and objective analysis of colorimetric data without the need of complicated interpretation data methods, and consequently it represents a significant milestone in the current and future development of ePADs for clinical diagnostics

Finally, attracting increased attention from the research community is the development of mobile device-based healthcare as a revolutionary approach for monitoring medical conditions. More systems have computerized traditional clinical tests to design functions and interactions of smartphone-based rehabilitation systems [103] regarding diseases like as stroke and cardiac failure [104]. Moreover, sophisticated platforms were developed for a better-targeted cancer therapy and improved follow-up care, to make the care process more effective in terms of clinical outcome. On the other hand, there is also the need to develop the μPAD for personalized toxicity studies. Therefore, the future trends on theranostic applications of the μPAD will be developed from the bench-to-bedside and updated to produce patient-friendly analytical assays [105].

## 6. Medical Clinical Applications

In cancer, the next generation of POC will probably be represented by 2-D material-based electrochemical biosensors/sensors [106] such as electrochemical apparatus [107], lateral flow assays (LFAs) [108], or paper-based colorimetric solutions [109]. The main biomarkers relevant in this field are nucleic acids such as mRNA and DNA; proteins such as antigens, enzymes, and peptides; some small molecules such as the reactive species of oxygen and nitrogen; and, notably, the protonation state [83]. Due to the chaotic nature of this medical condition, multiple marker solutions can fit the clinical needs better. Aptamers [110] are artificial oligonucleotides selected through a Systematic evolution of ligands by exponential enrichment (SELEX) procedure. They are gaining appreciation as ultra-specific, stable probes. Their use spans therapeutics to diagnostics, where they can be used as biomarker surrogates or in so-called aptahistochemistry, an evolution of immunohistochemistry [111].

In this section of the review some examples related to the more diffuse cancer types and infectious diseases are reported. Cancer-related applications are summarized in Table 1, infectious disease applications are summarized in Table 2.

In breast cancer aptamers are gaining momentum, and they find usage in therapy, as well as in the detection of diagnostic and prognostic markers such as aberrant HER-2 forms [112,113], α-estrogen receptor status [114], vascular endothelial growth factor (VEGF) [115], osteopontin [116], Michigan Cancer Foundation-7 (MCF-7) cells [117], anterior gradient homolog 2 (AGR-2) protein [118].

In lung cancer, one of the main issues is early diagnostics and effective screening. The problem is to find a proper balance between sensitivity and specificity, and complexity, time, and expenses. According to a recent review by Roointan and colleagues [119], the main biosensor approaches in lung cancer early diagnosis are electrochemical, optical, and piezoelectric (mass-based). The best clinical, analytical, and technological performances are achieved by electrochemical sensors. The main biomarkers sensed by those instruments, are: VEGF165 [120,121], EGFR [122], Annexin II and MUC5AC [123], HIF-1α [124], NADH levels [125].

In colorectal cancer (CRC) the main early diagnostic biomarker is the fecal occult blood test (FOBT) [126,127]. The most advanced FOBT tests commonly available on the market are immunochemical [128,129]. These can be further divided into qualitative and quantitative [130]. One of the main advantages of the immunological-based approach is that patients are allowed to stay on a regular diet without the need to stop drugs known to interfere with the guaiac-based FOBT [128].

Instead of looking for occult blood in stools, it is possible to approach early diagnosis looking for common genetic aberrations commonly found in CRC, such as K-Ras, adenoma polyposis coli (APC), p53, and microsatellite instability. Moreover, the novel DNA tests comprise epigenetic analysis of methylated genes for vimentin, secreted frizzled-related protein 2 (*SFRP2*), bone morphogenetic protein 3 (*BMP3*), N-Myc downstream-regulated gene 4 protein (*NDRG4*), and tissue factor pathway inhibitor 2 (*TFPI2*), using as analytical matrix feces or venous blood. Another promising target for early diagnosis and screening are fecal miRNAs [129,130,131,132,133,134,135,136,137,138].

A totally disruptive approach to FOBT is an ingestible micro-bio-electronic device (IMBED) based on environmentally resilient biosensor bacteria for in situ biomolecular detection, coupled with miniaturized luminescence readout electronics that wirelessly communicate with an external device. According to the authors, gut biomolecular monitoring could be more precise and faster than any other laboratory methods [139].

Another biomarker relevant in GI tract neoplasms is sarcosine. It is not only associated with CRC and stomach cancer, but also with prostate cancer and neurodegenerative disorders. Analytically it is relevant to human pathology in the food and fermentation industry. Many biosensor-based approaches have been tried to measure this analite: amperometric biosensors, potentiometric sarcosine biosensors, impedimetric sarcosine biosensors, photoelectrochemical (PEC) biosensors, and immunobiosensors. All these methods have been recently reviewed by Pundir at al. [140].

In the infectious disease market, many different tests received the Clinical Laboratory Improvement Amendments (CLIA) waivers that enable POC use [141]. Most POC rapid tests use lateral flow immunoassay (LFIA) technology, a limited number of POC diagnostics utilize molecular approaches. One of the most interesting molecular methods is nicking enzyme amplification reaction (NEAR) [142], an isothermal nucleic acid amplification. Back in 2015, the FDA approved the Alere i influenza A & B test [143] based on NEAR technology, which is an isothermal DNA amplification technique. Later also a test for group A Streptococcus (GAS) that uses throat swabs as samples was CLIA waived [144], and one for respiratory syncytial virus (RSV) [145].

Recently some reviews summarized monographically the state of the art in POC testing of common conditions. Kozel and collaborators [141] provided data for cryptococcal antigen meningitis and malaria. Grebely and colleagues in 2017 reviewed the offer for HCV (Hepatitis C Virus) POC testing [146]; Gaydos et al. *Trichomonas vaginalis* [147]; Hurt et al. HIV (Human Immunodeficiency Virus) [148]; Kelly and his group *Chlamydia trachomatis* [149]; Basile and collaborators in 2018 reviewed POC testing for respiratory viruses [150]; and Nzulu and colleagues in 2019 the one for *Leishmania* [151].

Concerning other experimental techniques in infection diagnostics and specific biosensor approaches, refer to the work by Datta et al. [152].

There are two other major chapters in infectious disease relevant for this review, apart from pure diagnosis: antibiotic susceptibility testing [153,154] and sepsis early diagnosis [155,156,157,158].

**Table 1 micromachines-11-00123-t001:** Clinical application in oncology.

Device	Disease	Reference
RNA aptamers anti HER2	Breast cancer	[110]
Nanotube-wrapped anti-HER2 protein aptamers	Breast cancer	[113]
DNA aptamer anti ERα	Breast cancer	[114]
Upconversion nanoparticles DNA aptasensor for VEGF	Breast cancer	[115]
Electrochemical aptasensor for osteopontin	Breast cancer	[116]
MCF-7 aptamer-functionalized magnetic beads and quantum dots based nano-bio-probes	Breast cancer	[117]
G-quadruplex structured DNA aptamer against AGR-2	Breast cancer	[118]
Electrochemical detection of VEGF based on Au–Pd alloy-assisted aptasensor	Lung cancer	[120]
Electrochemical aptasensor for VEGF	Lung cancer	[121]
EGFR DNA sandwich-type electrochemical biosensor	Lung cancer	[122]
Amperometric immunosensors for Annexin II and MUC5AC	Lung cancer	[123]
Amperometric sensing of HIF1α	Lung cancer	[124]
Amperometric sensor for NADH (nicotinamide adenine dinucleotide) using activated graphene oxide	Lung cancer	[125]
Volatile organic compounds (VOC) (Different methods)	GI (GastroIntestinal)-tract cancer	[126]
Pyruvate kinase isoenzyme type M2 (M2-PK) (Different methods)	GI-tract cancer	[127]
Ingestible micro-bio-electronic device (IMBED) and miniaturized luminescence readout electronics, wirelessly communicating with an external device	GI-tract cancer	[139]
Sarcosine (Different methods)	Colorectal, prostate, and stomach cancer; Alzheimer, dementia, sarcosinemia	[140]

**Table 2 micromachines-11-00123-t002:** Clinical application in infectious diseases.

Device	Disease	Reference
Isothermal nucleic acid amplification test	Influenza	[142,143]
Isothermal nucleic acid amplification test	Group A beta-hemolytic streptococcus	[142,144]
Isothermal nucleic acid amplification test	Respiratory syncytial virus	[141,145]
Different methods (review)	Hepatitis C	[146]
Different methods (review)	*Trichomonas vaginalis*	[147]
Different methods (review)	HIV	[148]
Different methods (review)	Urogenital *Chlamydia trachomatis*	[149]
Different methods (review)	Viral respiratory tract infections	[150]
Loop-mediated isothermal amplification (LAMP)	*Leishmania* spp	[151]
Gene Specific DNA Sensors	Pathogenic Infections	[152,154]
Different methods (review)	Antibiotic-Susceptibility Profiling	[153,154]
Different methods (review)	Sepsis (Lactate)	[155,156]
Microfluidic biochip	Sepsis (CD64)	[157,158]

## 7. Conclusions

The size of the global market for sensors is expected to increase in the near future due to the (i) growing demand for devices that meet the needs of early and reliable analysis of diseases and (ii) the fast pace at which technology is developing, providing products that satisfy those needs, with the additional characteristics of low energy consumption, eye-catching design, and ease of use. Here, we have reported on the most up-to-date nanotechnology prototypes in the field of sensor devices. Despite the frontier technology many of these devices possess, their functionality is often compromised by a lack of care and strategy in the pre-analytical and device-maintenance phases. In this context it is important, for example, to identify the optimal operation-interval of the sensor and tune this interval such that it matches with the range of values of the biological sample signal that are clinically relevant. Such a range, in turn, depends on the characteristics of the biological sample, or matrix (tissue position and blood/urine collection), and on the degree of advancement of the disease. This is to say that the optimization and use of a device in biomedicine is not simple and requires cooperation across disciplines and a multidisciplinary approach to ensure the right technology is adopted in the right conditions and at the right time. The many interesting contributions to biomedical nanoelectronics that we have reviewed in this paper are examples of how technology has developed to meet the needs of medicine. The nature of the problems that a new technology has to face is bifold: on one side, it has to solve specific scientific problems, and on the other side it has to adapt such a solution at the interface with medicine. Thus, it is likely that the evolution of the field of nanoelectronics in the near future will be guided by the definition of new problems in medicine.

## Figures and Tables

**Figure 1 micromachines-11-00123-f001:**
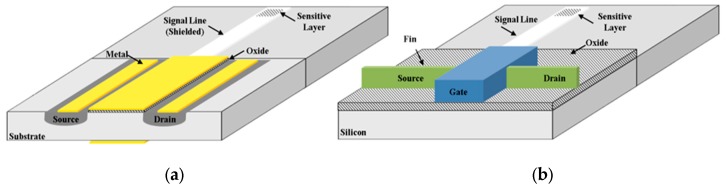
Representative view of a device based on extended gate field effect transistor (EGFET) (**a**) and (**b**) fin field-effect transistor (Fin-FET) technology (not in scale).

**Figure 2 micromachines-11-00123-f002:**
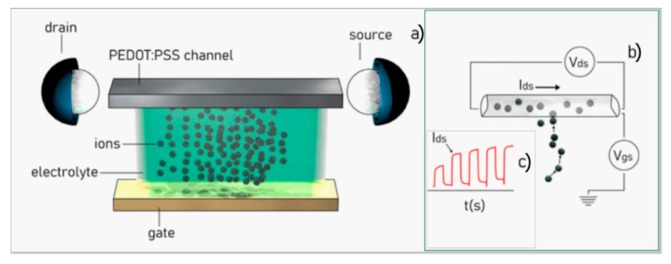
Scheme of a conventional organic electrochemical transistor (OECT) device, in which a solution is connected to the device through the gate, source and gate electrodes, and a conductive polymer channel (**a**). Upon application of an external voltage at the gate and the drain (**b**), a current of ions flows to the source generating a continuous function of time (**c**).

**Figure 3 micromachines-11-00123-f003:**
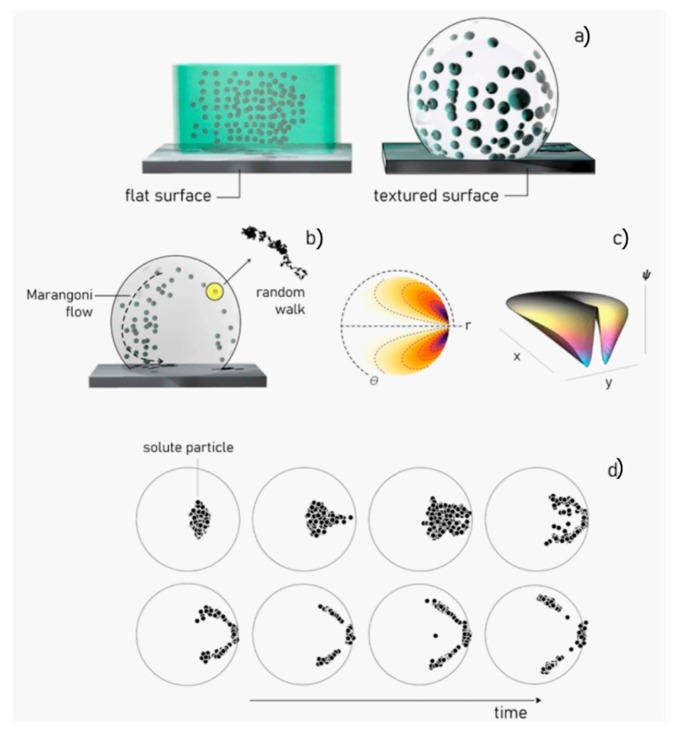
Nanoscale modification of a surface can make that surface super-hydrophobic (**a**). In a drop on a super-hydrophobic surface, the motion of particles is determined by the combination of diffusion and Marangoni convective flows (**b**). The inset reports a graphical representation of the potential functions ψ that describe the velocity field within a spherical drop (**c**). The displacement of particles in a drop can be determined from the velocity field using the Langevin equation and a numerical scheme (**d**).

**Figure 4 micromachines-11-00123-f004:**
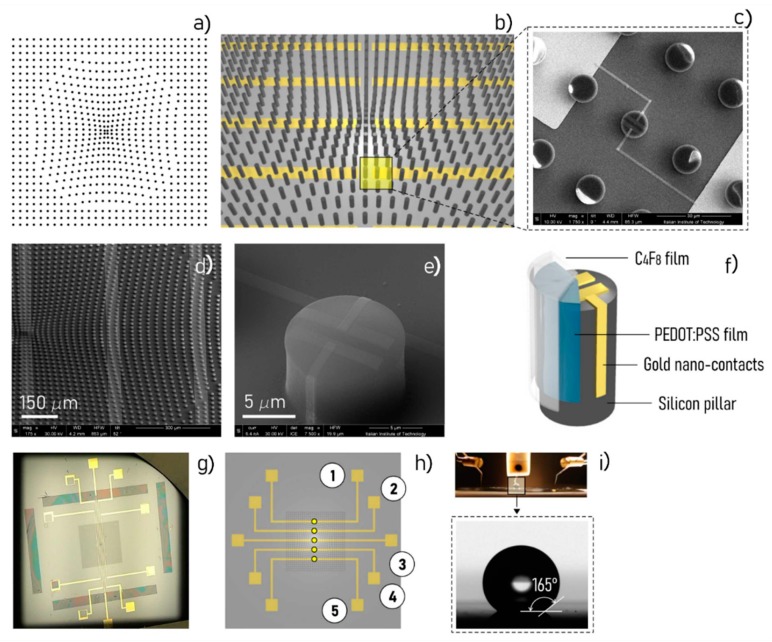
Surface enhanced organic electrochemical transistors (SeOECT) devices are based on a non-periodic array of micro-pillars (**a**). Some of those pillars are contacted through circuits to an external electric probing station (**b**) and are equipped with nanoelectrodes for site selective measurement of the ionic current: each of those pillars is named sensor (**c**–**f**). The device imaged with a camera lens, the distance between the parallel gold circuits is 1 cm (**g**). The sensors are placed in line on the device, symmetrically with respect to the center of the device (**h**). Due to the characteristics of the substrate, during operation the liquid sample maintains a spherical shape (**i**).

**Figure 5 micromachines-11-00123-f005:**
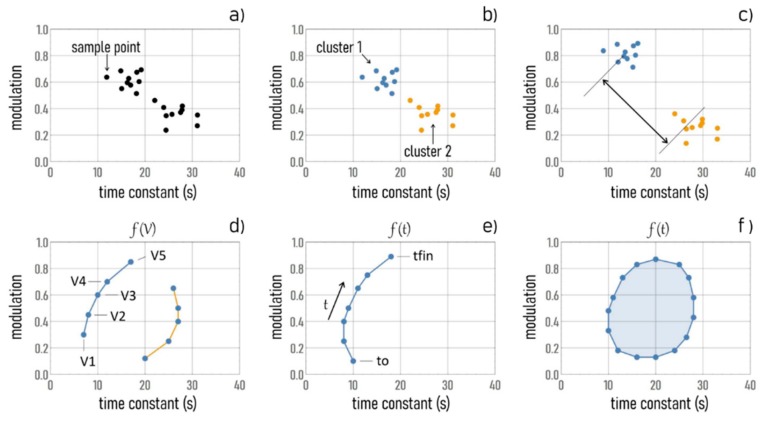
The response of the SeOECT devices is described by the sole variables modulation and time constant (**a**). The scatter plot of modulation against time constant may be indicative of differences between samples and can be used to operate sample separation, clustering, and classification (**b**). Separation between species can be improved by setting high voltage values and using sensors positioned at the border of the drop, where Marangoni flows are maximized (**c**). The modulation and time constant variables can be parametrized by voltage (**d**) and by time (**e**). The form of these trajectories can be indicative of the time evolution of the system (**f**).

**Figure 6 micromachines-11-00123-f006:**
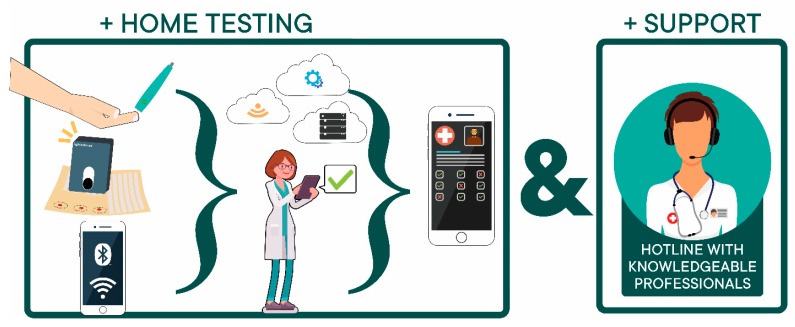
Flowchart of the use of an Android smartphone for image processing and paper-based devices. In this example, the test is performed at home, the data collected from the paper-based device through the Android application, and adequate software is sent to qualified medical personnel to support the management of the results obtained.

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
