# Peer review of "Emerging Designs of Electronic Devices in Biomedicine"

_micromachines, 2020, doi:10.3390/mi11020123_

Round 1
Reviewer 1 Report
The paper entitled “Emerging designs of electronic devices to enhance heterostructures and unique functionalities” from Coluccio et al. is a review of new transistor-based devices that can be fruitfully used to build LOC for biomedical applications.
For sure, the topic is quite interesting and new, even if different reviews have been presented in the last years. The language and style should be improved, especially in some parts that are worse than others and not completely clear. A few misprints and some parts that must be rephrased to be understandable. At the same time, some parts are not presented in a “review form” but they are based on one or two papers presented by the same authors of the review. At least one section is mainly based on a single paper, as reported below. Sometimes, it is really hard to understand what the authors would like to highlight from the literature. Maybe they should add figures and tables to make the review more practical and clear to the audience.
Therefore, I think the paper cannot be published in Micromachines in this form. In order to improve the quality of the paper, I listed a few points the authors should address before acceptance:
- Take care with acronyms. They should be defined the first time you insert them in the manuscript. Some are not defined. At the same time, they are not really useful if used only once in the manuscript. Please, check the whole paper.
- A lot of misprints and mistakes are present, so I suggest to read the paper carefully and proofread it. Maybe a native speaker is required. A Few examples (not exaustive!!!): lines 114 (check the verb), 151 (not clear), 179 (the line start with a title, or what?), caption of Fig. 3, line 249 (“transitions”?), line 203 (“reading”?), line 466 (“aptameres”), etc, etc…
- Section 4 is mainly based on a single paper published by the authors. This should not be allowed for a review paper. Please, the section is quite interesting but it should be based also on other works presented by other authors!! If, as I think, these kind of sensors are only made by the authors, they should try to combine these sensors in the OECT family, reducing the self-reporting. The same for section 5, especially between lines 369 and 388.
- There are other kind of emerging FET that the authors didn’t reported, such as GFET (Graphene FET), even if they are quite often reported in the literature, especially for biosensing applications. Please, add something on the other emerging electrical/electrochemical sensors present in the literature.
- Is there any kind of commercial sensor based on the technology reported by the authors? This should be another interesting point to be explored by the authors.
- As reported above, a table should be added in section 5 and 6. For section 6 it is really needed because the reported examples are kind of hidden in the text.
- Line 370, the authors stated that they produced the “first example of LOC for cellular biomarker” but I think this sentence is too much…Please revise it.
- Page 12 is really cumbersome and not well integrated into the review. If you want to add something related to the future trends in sensing by using smartphone and related devices you should make a new section. A section on the future trends of the presented technology should be of great impact.
- In section 6, you should insert a table and avoid the titles such as “breast cancer” at line 470.
- Lines 516-519 should be expanded as well. The authors spent a lot of words to describe other non really relevant aspects of FET-based sensors, but they only put a few words on relevant parts.
- The conclusions are not real conclusions. Please, write ita gain.
Author Response
Dear Editor,
We are pleased to submit to Micromachines, a revised version of the manuscript entitled “Emerging designs of electronic devices in biomedicine”, and a detailed response to the comments of the Reviewers. The MS has been revised to comply with the observations of the Reviewers. Specifically:
We have corrected typos everywhere in the MS and we improved English and style of the MS. Where necessary, we have rewritten words or sentences to convey more clearly the message of the paper. Acronyms have been explained everywhere in the text. The Abstract, the Introduction and the Conclusions of the paper have been reformulated. The Title itself has been changed as “Emerging designs of electronic devices in biomedicine”. In response to the observation of the first Reviewer, the section entitled “Organic electrochemical transistors devices” has been expanded to include additional examples, additional applications and case studies pertaining organic electrochemical transistors (OECTs).
Considering the corrections made to the manuscript, and the positive comments of all the Reviewers on the work, we hope that the paper can be accepted for publication in this present form. We warmly thank the Reviewers that, with their comments, contributed to improve the work.
In the following FILE, you will find the original comments from the reviewers (bold black text) and the point-by-point response of the authors (bold blue text). In the manuscript, new included portions are highlighted in YELLOW, whereas the text removed from the original version is highlighted in GRAY.
With Regards
The authors

Reviewer 2 Report
Dear authors,
thank you for your interesting work.
Some small comments:
line 28: at now not sufficiently(!) known or which need (!) a personalized Approach.
line 43: this review aims at Focusing on an
line 44: and bioelectronics devices. (sentence end) The circuit
line 47: recovery possibility, they are (comma!)
line 48: an electrical Signal (!) to detect
line 55: examples are reported here.
line 66: the gate consists of an(!) SiO2 layer
line 78: FinFETs resulted in an attractive option for the fabricatoion of a(!) non-planar
line 80: which guarantees immunity
line 83: the OTFTs' (!)
line 84: of carrier mobility, (!) operating frequency
line 87: biosensor's (!)….is a(!) standard
line 88: which ensures(!)
line 90: as follows: (!)
line 92: Please use subscript V_ref V_ds
line 93: V_th ….voltage, (!) which is related
line 95: the field-effect device.(!) {sentence end} E_ref(!)
Fig 1: please write a), b) instead of solely a, b - applies to all figures
line 102: A higher transconductance (!) is desirable ...it results (!)
line 103: as a lower current,(!) a lower V_th
line 106: biosensors(!)
line 110: and thus it is (!) mature for
line 111: biosensors(!)
line 113: other approaches
line 114: (TFET) are(!) one of the most recent devices, (!) which base
line 127: the development(!) time
line 128: <better place on next page>
line 129: the role of organic (lower case)
line 146: plants [40] and(!) as a sensor
fig 2: a), b), c) ---place fig c) a little bit more apart
line 179: Nanotechnology ...of OECTs: (double bullet!)
fig. 3: a), b), ...
line 216, 217: A Sketch defining d, Delta and h would help
line 252: distance -> radius
line 259: They are placed symetrically with(!) respect to
line 279: Layer A: (double bullet)
line 284: Layer B: (double bullet)
line 291: consists of(!)
line 296: consists of(!)
line 310: "the" twice
line 320: SeOECTs: (double bullet)
fig. 5: a), b)
please provide dimensions, e.g. time constant [ns], modulation [?]
line 372: "it is described" - sentence order is corrupt
line 400: the chosen detection method
line 401: depend on(!)
line 424: and (!) paper (reduce space)
line 429: will be send
line 466: can fit the clinical needs better.
"oligos" - what does it mean?
line 470: Breast Cancer: (Double bullet)
line 473: Lung Cancer: (double bullet)
line 523: In(!) this review
line 527: About the functionality (sentence order corrupt)
line 531: is the choice (!)
Author contributions - not answered!
Funding?
references do: [1], [2], ...
Author Response

(The authors gave the same response as above.)

Round 2
Reviewer 1 Report
I would like to thank the authors because they try to fullfil all the request asked at the first round. Anyway there are some missed points:
- The language should be ckecked in all the manuscript, not only in some section. Please, carefully proofread the paper that still contains errors (e.g. “an android smartphones” or line 506 “aptameres”). Please, revise also the substript/superscript when needed (e.g. “O2” at line 416?).
- The authors mislead the comment on acronyms. The acronyms should be defined the first time only, and not each time. At the same time, avoid an acronym if it is used only once in the text. By the way some of them are still missing (e.g. “GI” is not explainded) . So please, revise the text;
- Line 179, now 221 was not changed and seems meaningless. Why the authors started with “Nanotechnology formulations of OECTs: In the scheme presented above...”? Please revise!!
- Caption of Figure 2 is still not clear. Check the language, please.
- Line 289? Check the language, please.
- Line 236 was not changed...
- Section 4 on seOECTs is still based on a single paper published by the authors!!! This is not allowed for a review paper! Please, the authors should revise this section, because in this form is too much self-referencing for a review. Then, this section should be based also on other works presented by other authors. The authors, instead of combining this section with the previous one, reducing it to avoid plagiarism with respect to their papers, they increased section 3 (increasing also the self-referencing), but maintaining the problem related to section 4 that was not changed at all. At the same time the self-referencing is around 20%, so please be careful.
- The same for section 5, especially between lines 409 and 427. The authors should reduce the self-reporting from their papers.
- The part extrapolated from section 5 (now section 6) should be really revised and not only separated in a new section. This part is really cumbersome and not well integrated into the review. The new section is fine, but the text should be revised as well to integrate it into the review. Especially between lines 462-465, 475-482.
- The new section 6 (page 13) is still really cumbersome and not well integrated into the review. It is fine that you create the new section as suggested, but also the text must be revised because it is not really integrated into the review context!!! Please, revise it.
- The authors reply that there is “A commercial graphene-based biosensor is the Agile Biosensor Chip – NTA, used overall for research purposes, allowing the immobilization of recombinant proteins.” They should add this to the paper. Maybe where they talk about GFET. Please, revise it.
Author Response
Paper: Manuscript ID: micromachines-669571
Title: Emerging designs of electronic devices in biomedicine
Dear Editor,
We are pleased to submit to Micromachines, a revised version of the manuscript entitled “Emerging designs of electronic devices in biomedicine”, and a detailed response to the comments of the Reviewers. The MS has been revised to comply with the observations of the Reviewer. Specifically:
We have corrected typos everywhere in the MS and we improved English and style of the MS, writing newly words or sentences for a better comprehension of the text. Acronyms have been explained everywhere in the text and defined the first time only. In response to the observation of the first Reviewer, sections 4, 5, 6 have been revised eliminating not relevant information and rewriting the not clear text.
Considering the corrections made to the manuscript, and the positive comments of all the Reviewers on the work, we hope that the paper can be accepted for publication in this present form. We warmly thank the Reviewers that, with their comments, contributed to improve the work.
In the following, you will find the original comments from the reviewers (bold black text) and the point-by-point response of the authors (bold blue text). In the manuscript, new included portions are highlighted in YELLOW, whereas the text removed from the original version is highlighted in GRAY.
With Regards
The authors
Reviewer 1
I would like to thank the authors because they try to fullfil all the request asked at the first round. Anyway there are some missed points:
- The language should be ckecked in all the manuscript, not only in some section. Please, carefully proofread the paper that still contains errors (e.g. “an android smartphones” or line 506 “aptameres”). Please, revise also the substript/superscript when needed (e.g. “O2” at line 416?).
The language has been revised in all the manuscript.
- The authors mislead the comment on acronyms. The acronyms should be defined the first time only, and not each time. At the same time, avoid an acronym if it is used only once in the text. By the way some of them are still missing (e.g. “GI” is not explainded) . So please, revise the text;
All the acronyms in the MS have been checked and defined only the first time, avoiding repetitions.
- Line 179, now 221 was not changed and seems meaningless. Why the authors started with “Nanotechnology formulations of OECTs: In the scheme presented above...”? Please revise!!
The phrase was reformulated to make more fluid and clear the writing.
- Caption of Figure 2 is still not clear. Check the language, please.
Caption of Figure 2 has been revised as follows:
“Nanoscale modification of a surface can make that surface to be super-hydrophobic (a). In a drop on a super-hydrophobic surface, the motion of particles is determined by the combination of diffusion and Marangoni convective flows (b). The inset reports a graphical representation of the potential functions that describe the velocity field within a spherical drop (c). The displacement of particles in a drop can be determined from the velocity field using the Langevin equation and a numerical scheme (d).”
- Line 289? Check the language, please.
The correspondent paragraph has been revised and some technical details, considered not relevant for the review, eliminated, comprising line 289.
- Line 236 was not changed...
Line 236 has been revised as follows:
“are derived as the derivatives of the stream functions ψ(r,θ) with respect to the coordinates r and θ. The stream function are potential functions that describe the characteristics of a fluid flow, they have been originally derived by Tam and collaborators and [70] read:”
- Section 4 on seOECTs is still based on a single paper published by the authors!!! This is not allowed for a review paper! Please, the authors should revise this section, because in this form is too much self-referencing for a review. Then, this section should be based also on other works presented by other authors. The authors, instead of combining this section with the previous one, reducing it to avoid plagiarism with respect to their papers, they increased section 3 (increasing also the self-referencing), but maintaining the problem related to section 4 that was not changed at all. At the same time the self-referencing is around 20%, so please be careful.
We thank the Reviewer for the observation. Following suggestions from the Reviewer, we have removed many of the self-references from sections 3 and 4 (more than 12), we have included additional references and commented on additional papers from other research groups either in relation to the biomedical applications of OECTs, and regarding the description of super-hydrophobic surfaces. Moreover, we have merged sections 3 and 4 together and reduced much of the content of the original section 4 on SeOECTs. In the revised version of the paper, results pertaining the SeOECTs are not desegregated from the rest of the section on OECTs.
- The same for section 5, especially between lines 409 and 427. The authors should reduce the self-reporting from their papers.
- The part extrapolated from section 5 (now section 6) should be really revised and not only separated in a new section. This part is really cumbersome and not well integrated into the review. The new section is fine, but the text should be revised as well to integrate it into the review. Especially between lines 462-465, 475-482.
- The new section 6 (page 13) is still really cumbersome and not well integrated into the review. It is fine that you create the new section as suggested, but also the text must be revised because it is not really integrated into the review context!!! Please, revise it.
We thank the Reviewer for the observations. Following suggestions from the Reviewer, we have (i) reduced the self-reporting from our papers and (ii) revised section 5 and 6 (in the new revision are section 4 and 5) to make the text more flowing and clear.
- The authors reply that there is “A commercial graphene-based biosensor is the Agile Biosensor Chip – NTA, used overall for research purposes, allowing the immobilization of recombinant proteins.” They should add this to the paper. Maybe where they talk about GFET. Please, revise it.
The phrase was added in the text.

Round 3
Reviewer 1 Report
I would like to thank the authors because they fullfil most of the request asked at the previous round. I think now the review has an increased value and then the paper can be published in Micromachines.